# Diet and density effects on growth, proximate composition and fatty-acid profile of *Hediste diversicolor* juveniles

Zabihollah Pajand[1,2]*, Ayoub Yousefi[1], Hamed Abdollahpour[3,4], Esmaeil Hosseinnia[1], Arash Lebria[1], Younes Golalipour[1], Parham Pajand[2], Naghmeh Jafari Pastaki[3,4], Mohebali Pourgholam[1]

1 International Sturgeon Research Institute, Iranian Fisheries Science Research Institute, Agricultural Research Education and Extension Organization (AREEO), Rasht, Guilan, Iran, 2 Zist Palayeshgar Khazar Company (ZPK), Guilan Science and Technology, Rasht, Guilan, Iran, 3 Comparative Molecular and Integrative Biology, Centro de Ciências do Mar, Universidade do Algarve, Faro, Portugal, 4 Fisheries Department, Faculty of Natural Resources, University of Guilan, Sowmeh Sara, Guilan, Iran

* zpajand@gmail.com

## Abstract

This study evaluated the nutritional quality of *Hediste diversicolor* cultivated using waste from beluga sturgeon (*Huso huso*) aquaculture. Over a 60-day trial, polychaetes were reared at four stocking densities (1000–7500 individuals per square meter (ind./m²)) and fed either commercial sturgeon feed or processed farming waste. Results demonstrated that sturgeon waste effectively supported polychaete culture, with survival rates ranging from 53% to 80%. However, worms fed commercial feed exhibited superior growth, yielding over three times greater biomass than waste-fed groups. Nutritional composition was significantly affected by both diet and density. Worms fed commercial feed at an intermediate density (2500 ind./m²) achieved the highest protein content (9.23%) and most beneficial fatty acid profile, characterized by a high PUFA content (46.85%) and n-3/n-6 ratio (0.39). While waste-fed worms showed reduced nutritional quality, an intermediate density of 2500 ind./m² also optimized their composition. A comparative analysis confirmed the commercial feed was richer in protein and essential fatty acids (EPA+DHA) than the processed waste. The study confirms the viability of using sturgeon waste to produce polychaete biomass for aquaculture. However, nutritional trade-offs suggest supplemental feeding may be required to achieve premium quality for aquaculture feed, providing critical benchmarks for implementing circular economy principles in aquaculture systems.

## Introduction

The conservation of critically endangered sturgeon stocks in the southern Caspian Sea basin has elevated commercial farming from a mere commercial activity to an

**Data availability statement:** All data generated or analysed during this study are included in this article [and its supplementary information files]. Specifically, the complete dataset is provided in S1 Dataset (Supplementary_material.xlsx).

**Funding:** This study was financially supported by grants from the Iran National Science Foundation (INSF Contract no. 91002674) program project. The funders had no role in study design, data collection and analysis, decision to publish, or preparation of the manuscript.

**Competing interests:** The authors declare that they have no competing interests. Specifically, none of the authors has any financial interests nor any non-financial interests that could be perceived to influence the work reported in this paper.

indispensable tool for protecting global biodiversity and ensuring the survival of these ancient species [1,2]. This strategic pivot is a direct response to the catastrophic decline of wild populations, which face existential threats from relentless overfishing, illegal harvesting, rampant habitat degradation, and the systematic destruction of natural spawning grounds. Consequently, global aquaculture has expanded sturgeon farming, recognizing its dual role in supplying high-value products such as caviar and meat while supporting urgent conservation goals [3–5]. However, this rapid expansion poses a paradoxical environmental challenge, as intensive coastal farms continuously release nutrient-rich effluents into marine waters. These discharges, largely uneaten feed, fecal matter, and nitrogenous byproducts such as ammonia, contain elevated levels of organic carbon, nitrogen, and phosphorus [6,7]. When discharged without treatment, these substances act as strong pollutants that accelerate eutrophication in coastal waters. This process promotes harmful algal blooms, hypoxia, and damages benthic habitats, ultimately undermining the ecosystems essential for sturgeon conservation. Addressing this issue requires the establishment of stricter regulatory frameworks grounded in rigorous environmental standards to maintain pollutant loads within acceptable limits. The primary challenge for contemporary aquaculture is therefore to reconcile rapid productivity growth with the maintenance of environmental [8,9].

In response to this challenge, the industry has shifted from costly waste treatment to innovative waste valorization strategies. This approach promotes a circular economy in aquaculture, using the waste from one species as a resource for another. The concept of Integrated Multi-Trophic Aquaculture (IMTA) exemplifies this approach, emphasizing the critical need for biological technologies that not only mitigate environmental impacts but also create tangible economic value from waste streams, thereby transforming a traditional cost centre into a potential revenue opportunity [10,11]. Within this framework, the polychaete worm *Hediste diversicolor* (formerly *Nereis diversicolor*) emerges as a premier candidate for aquaculture, owing to its exceptional ecological and economic attributes. Its value is derived from a remarkable dual functionality. Primarily, it is a highly efficient detritivore, with an extensively documented ability to consume and break down organic waste, including particulate fecal matter and surplus feed, thereby directly reducing the organic load and nutrient concentration of aquacultural effluents [12–16]. Secondly, it is a nutritionally rich organism. Its high protein content and excellent profile of essential fatty acids render it a highly valuable live or processed feed for a variety of commercially important species, including various penaeid shrimp (*Penaeus kerathurus*, *Penaeus vannamei*) and juvenile sturgeons [17–21]. So, it creates a highly efficient circular economy loop, where aquaculture waste is converted directly into high-value biomass, closing the nutrient cycle.

Live prey remains a cornerstone of early larviculture because they deliver highly digestible protein, essential amino acids and conditionally essential long-chain n-3 highly unsaturated fatty acids (HUFA), and because their biochemical profile can be manipulated through feed and enrichment protocols to meet larval nutritional requirements. Studies on rotifers and Artemia demonstrate how different

commercial feeds, enrichers and microalgae substantially affect biochemical composition, survival, growth and HUFA content of live prey and therefore of the larvae that consume them [22–25]. To broaden the suite of available live feeds, cultured polychaetes (e.g., *H. diversicolor*) have been proposed and trialed as an alternative: polychaetes provide high protein content, substantial lipid and PUFA fractions, and their EPA/DHA content can be increased by short-term dietary enrichment, making them attractive both as direct feed and as a vehicle for delivering specific fatty acids to larvae [26]. Below, we therefore summarize fatty-acid composition and enrichment potential of polychaetes and discuss implications for their use in sturgeon effluent systems. Furthermore, the nutritional composition of the resulting worm biomass, largely determined by diet, is crucial when it is intended as live feed in hatchery or grow-out operations. Sturgeon farming waste is a nutritionally modified substrate, as effluents typically contain lower lipid and protein levels due to partial digestion by the fish. Recent molecular and physiological work shows that *H. diversicolor* possesses the enzymatic machinery (multiple Elovl elongases and front-end and methyl-end desaturases) required to biosynthesise long-chain PUFA (LC-PUFA) from C18 precursors and to modulate LC-PUFA output in response to diet and culture conditions. This contrasts with commonly used live prey (rotifers, *Artemia*), whose LC-PUFA content is principally determined by diet and enrichment (although some rotifer species show limited endogenous conversion), implying that *H. diversicolor* can act both as a converter of low-quality side-streams into nutritionally valuable LC-PUFA and as a more intrinsically flexible live feed a capacity that is influenced by temperature, salinity and nutritional factors (including micronutrients such as iron) [27,28].The concentration of long-chain polyunsaturated fatty acids (LC-PUFA), particularly eicosapentaenoic acid (EPA, 20:5n-3), docosahexaenoic acid (DHA, 22:6n-3) and arachidonic acid (ARA, 20:4n-6), is a primary indicator of live-feed and diet quality because different fish life stages have distinct quantitative and functional requirements for these fatty acids. Early larvae typically require diets rich in n-3 HUFA and marine phospholipids (larval diets with ~10% dry weight as n-3 HUFA-rich lipids have been proposed as desirable), whereas juveniles and grow-out fish need adequate EPA+DHA for sustained growth and somatic performance, and broodstock diets must supply appropriate EPA/DHA/ARA ratios to optimise egg quality and larval viability. Species-specific requirement estimates exist (for example, many marine larvae show explicit HUFA thresholds; grow-out salmonids show improved long-term performance when EPA+DHA exceed ~2–3% of total fatty acids), and dietary microalgae or enriched live feeds can fulfil these needs when appropriately formulated. These life-stage differences justify the present focus on both (i) the fatty-acid composition of *H. diversicolor* and (ii) how its FA profile (and the capacity to modulate it) matches the specific PUFA needs of larvae, growers and broodstock [29,30]. These omega-3 fatty acids are essential for supporting normal growth, neurological development, and stress resistance [31,32]. A diet of nutritionally depleted sturgeon waste could theoretically result in a worm with an inferior fatty acid profile, potentially undermining its efficacy and value as a live feed. Therefore, a comprehensive whole-body composition analysis, encompassing both proximate composition and detailed fatty acid profiling, is not merely beneficial but essential to truly evaluate the end product's quality and its suitability within the aquaculture value chain.

To address these gaps, this study is specifically designed to bridge these gaps by systematically evaluating the cultivation of *H. diversicolor* using effluent from beluga sturgeon (*Huso huso*) culture. Three targeted objectives guide our research: first, to assess the technical feasibility of utilizing raw sturgeon farm waste as a sole nutrient source for polychaete production; second, to identify the optimal stocking density that maximizes the outputs of rearing efficiency and sustainable worm production, measured through growth, survival, and total biomass gain; and third, to comprehensively evaluate the effect of a sturgeon waste diet on the worm's biochemical composition through detailed proximate analysis and fatty acid profiling to authoritatively determine its suitability as a high-value feed ingredient. By addressing these objectives, this research aims to provide a scientifically robust, and environmentally sustainable model for effluent management in sturgeon aquaculture, effectively transforming a waste product into a valuable resource and contributing to the principles of a circular economy as outlined by global sustainability objectives [33–35].

## Materials and methods

### Ethical information

Animal welfare, monitoring and endpoints. Juvenile beluga sturgeon used to produce aquaculture waste were maintained under the institutional IACUC oversight of the International Sturgeon Research Institute, Iranian Fisheries Science Research Institute, AREEO, Rasht, Guilan, Iran. The beluga sturgeon were not sampled for this study; they were used only as the source of aquaculture effluent that was processed into feed for the polychaete trials.

### Polyceate husbandry

*H. diversicolor* juveniles used in the experimental trials are marine invertebrates. Vertebrate-style humane endpoints for continued animals were therefore not applicable. Polychaetes were stocked at the densities reported in the experimental design and were observed twice daily (morning and late afternoon) for activity, feeding behaviour and signs of morbidity; water-quality parameters (temperature, salinity, dissolved oxygen) were recorded daily. Any moribund or dead individuals were removed immediately and recorded. Terminal sampling of polychaetes for proximate composition and fatty-acid analysis was performed at the scheduled end of the 60-day culture period; terminal sacrifice was performed by rapid cooling/immersion in an ice slurry followed immediately by freezing (–80°C) to preserve biochemical integrity. All personnel performing animal husbandry and sampling had documented training in aquarium husbandry, species-specific handling and the approved sampling procedures.

### Experimental animals and waste production system

This research was conducted at the International Sturgeon Research Institute (Sanger, Gilan Province, Iran). Both juvenile beluga sturgeon and larval *H. diversicolor* polychaetes were sourced from the institute's aquaculture facilities. The experimental setup utilized 60 juvenile beluga sturgeon with a mean initial weight of $42.9 \pm 3.5$ g. The fish were randomly distributed across twelve 100-L tanks at a stocking density of 5 fish per tank (equivalent to $1.4 \pm 0.2$ kg/m³). Throughout the 60-day experimental period, fish were fed a commercial extruded diet (Faradaneh Co., GTF1; 2 mm pellet size) at a rate of 3% of body weight per day, divided into four daily rations. The diet composition was as follows: 38–42% crude protein, 13–17% crude fat, 2–4% crude fibre, 7–11% ash, 5–11% moisture, and 1–1.5% phosphorus. A continuous flow-through system was maintained using well water at a rate of 2.5–3.0 L/min, with continuous aeration provided via air stones to ensure dissolved oxygen levels remained near saturation. Aquaculture waste, comprising uneaten feed and fecal matter, was collected daily via siphoning from the sturgeon tanks. The collected effluent was filtered through a 100-μm mesh plankton net to concentrate solid waste, which was subsequently oven-dried at 55°C for 24 hours. This dried sturgeon waste product was then used as the exclusive feed source for *H. diversicolor* in the subsequent polychaete culture trials.

### Polychaete culture and experimental design

*H. diversicolor* juveniles with a mean initial weight of $0.02 \pm 0.01$ g were obtained from the Caspian Bioremediation Knowledge-Based Company (Guilan, Iran). The worms were systematically divided into two primary dietary treatment groups: one group received a commercial sturgeon feed (C; Faradaneh Co. GTF1; 2 mm pellet size, 38−42% crude protein, 13−17% crude fat, 2−4% crude fibre, 7−11% ash, 5−11% moisture, and 1–1.5% phosphorus), while the other was fed waste recovered from beluga aquaculture operations (W). Each dietary regime was further evaluated across four stocking density treatments 1000, 2500, 5000, and 7500 worms per square meter, resulting in a total of eight experimental groups (C1, C2, C3, C4, and W1, W2, W3, W4). Fig 1 provides a schematic overview of the experimental design. The study utilized twenty-four 40-L tanks (60 × 40 × 20 cm), each equipped with a 5-cm layer of sterilized sand substrate collected from the Caspian Sea shoreline. The experimental setup followed a completely randomized design with three replicates per treatment group. Throughout the trial, polychaetes were exclusively fed their respective diets, with no supplemental

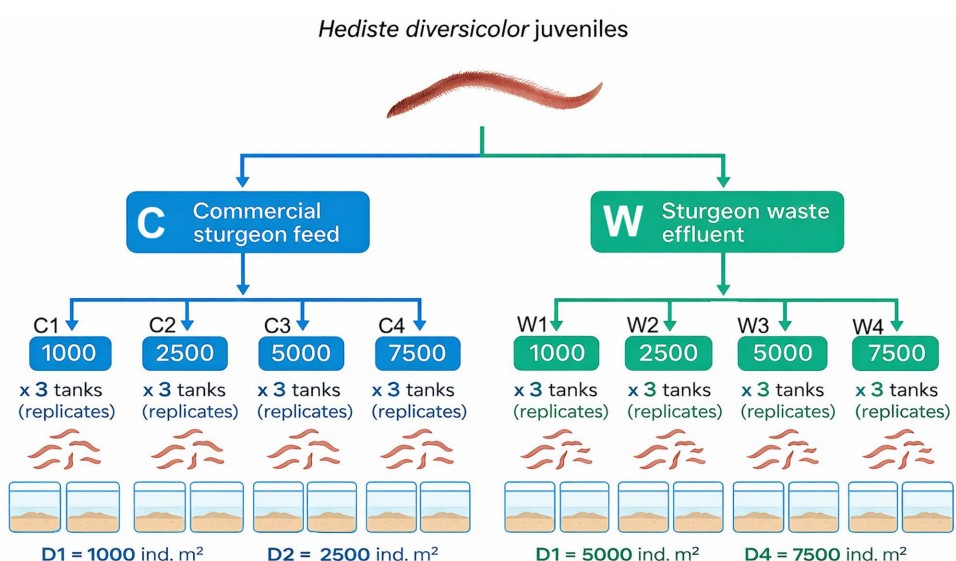

**Fig 1. Experimental design showing *Hediste diversicolor* juveniles reared under two dietary treatments (commercial sturgeon feed, C; sturgeon waste effluent, W) and four stocking densities (D1 = 1000, D2 = 2500, D3 = 5000, D4 = 7500 ind. m⁻²).** Each treatment–density combination was conducted in triplicate (three independent 40-L tanks) containing sterilized sand substrate.

nutrition provided. Polychaetes were fed at a daily ration equivalent to 1% of their total wet biomass, divided into two equal meals administered at 08:00 and 15:00 h. Feed amounts were adjusted periodically based on biomass estimates to maintain a constant feed-to-worm biomass ratio throughout the experimental period. To standardize the feeding regimen of the W group, waste effluent comprising uneaten feed and fecal matter was daily siphoned from the sturgeon culture tanks, processed, and allocated accordingly (Fig 1).

## Sampling and analytical framework

Following the 60-day culture period, a comprehensive sampling was conducted. All worms were recounted and weighed to finalize calculations of growth performance and survival rates. The experimental design facilitated a dual analysis: firstly, to characterize the physico-chemical properties of the effluent beluga after treatment by *H. diversicolor* across different densities, and secondly, to conduct a comparative nutritional assessment of the worms. The biochemical composition (focusing on crude protein and total lipids) and fatty acid profiles (particularly unsaturated fats) of worms fed sturgeon farming waste were rigorously compared against those fed the commercial reference diet.

## Calculation of performance indices and biochemical analysis

The optimal worm density was determined by evaluating key performance indices derived from the final biometric data as below:

Specific growth rate (SGR) (%/day) = 100 × (ln final weight) – (ln initial weight) / days
Weight gain (WG) (%) = 100 × (final weight) – (initial weight) / initial weight × 100
Daily growth rate (DGR) (g/day) = (final weight) – (initial weight) / days
Survival rate (SR) (%) = 100 × (final number of worms / initial number stocked)
Biomass gain (g/m²) = (final biomass) – (initial biomass)
Feed conversion ratio (FCR) = feed consumed / (final weight – initial weight)

Condition factor (CF) = 100 × body weight / total length³

Concurrently, water quality in the sturgeon tanks was maintained within strict parameters. A continuous flow-through system provided a constant inflow of 2.5–3.0 L/min of well water, with continuous aeration to sustain dissolved oxygen levels. Water flow rate was measured and verified using an inline ultrasonic flowmeter (ST501 Clamp-On Ultrasonic Flowmeter; Flo-Instru, Shenzhen, China) and was cross-checked weekly by volumetric measurement by collecting the tank outflow into a graduated cylinder for 60 s and converting the volume to L min$^{-1}$. Daily measurements using a HACH HQ40d multiparameter meters confirmed stable conditions, with mean values of 17.7 ± 0.6 °C for temperature, 8.0 ± 0.1 mg/L for dissolved oxygen, and 7.9 ± 0.1 for pH. Following biometric analysis, representative worm samples from each treatment were preserved and transferred to the laboratory for detailed biochemical analysis, including proximate composition and fatty acid profiling. The measurement of the physical and chemical factors of the water used for *H. diversicolor* cultivation revealed an average water temperature of 20.8 ± 1.1 °C, a dissolved oxygen concentration of 6.1 ± 0.2 mg/L, and a salinity of 12.8 ± 0.9.

### Sample preparation and proximate composition analysis

After collecting from the sediment, the worms are placed in clean running water with a salinity of 12.8 ± 0.9 for 24 hours to allow them to expel any fecal matter from their bodies. Upon conclusion of the trial, a representative sample of 15 g of *H. diversicolor* was randomly selected from each replicate tank, yielding a pooled sample of 45 g per group. These samples were immediately packaged, frozen at −20 °C, and transported to the laboratory for comprehensive biochemical analysis. Moisture content was determined gravimetrically by oven-drying samples at 105 °C for six hours until a constant weight was achieved. Crude protein content was estimated by quantifying total nitrogen using the Kjeldahl method and applying a conversion factor of 6.25. Total lipid content was determined by Soxhlet extraction using chloroform as the solvent [36]. Ash content was determined by incinerating the dried samples in a muffle furnace at 550 °C for 6 hours, and the remaining inorganic residue was weighed to calculate ash percentage.

### Fatty acid profiling by gas chromatography

Proximate composition and subsequent fatty acid analyses were conducted at the specialized Viromed laboratory (Rasht Industrial Town, Iran) under standardized analytical conditions. Total lipids for proximate composition were determined using the Soxhlet extraction method, providing quantitative lipid content for nutritional analysis. Fatty acid composition was determined using a two-stage analytical process. For fatty acid profiling, total lipids were extracted using the Folch method [37], which preserves individual fatty acids for subsequent analysis. The extracted lipids were then transesterified to fatty acid methyl esters (FAMEs) using the protocol described by Metcalfe and Schmitz [38]. FAME separation and identification were performed using a gas chromatograph (Young Lin 6500 GC, Korea) equipped with a flame ionization detector (FID) and a Supercowax-10 capillary column (60 m × 0.22 mm i.d. × 0.22 µm film thickness; Teknokroma, Spain). Using these two methods allows accurate determination of both total lipid content and detailed fatty acid composition from the same samples. The chromatographic separation utilized a programmed temperature gradient: initial temperature 90 °C (held for 0 min), increased to 170 °C at 25 °C/min, then to 190 °C at 2 °C/min (held for 4 min), further increased to 210 °C at 3 °C/min, and finally to 220 °C at 5 °C/min (held for 5 min). The injector and detector temperatures were maintained at 280 °C and 240 °C, respectively. Peak integration and quantification were performed using Varian Star Chromatography software (v. 6.41). Fatty acid concentrations were initially determined as mg fatty acid per g of wet tissue (mg/g) using an internal standard. For presentation, these absolute values were converted to percentages of total fatty acids (% of total FA) using the formula: % FA = (individual FA / sum of all quantified FA) × 100. This conversion allows direct comparison of fatty acid profiles across treatments and highlights compositional differences in SFA, MUFA, and PUFA, while absolute values provide information on tissue deposition. High-purity helium was used as the carrier gas, and individual FAMEs were identified by comparing their retention times with authenticated FAME standards, including C14:0 (myristic acid),

C16:0 (palmitic acid), C16:1 (palmitoleic acid), C18:0 (stearic acid), C18:1 (oleic acid), C18:2 (linoleic acid), C18:3 (linolenic acid), and C20:4 (arachidonic acid), obtained from Sigma-Aldrich, China. Extracted lipids were transesterified to fatty acid methyl esters (FAMEs) using the protocol described by Metcalfe and Schmitz [38], and FAME separation and identification were performed using a gas chromatograph (Young Lin 6500 GC, Korea) equipped with a flame ionization detector (FID) and a Supercowax-10 capillary column (60 m × 0.22 mm i.d. × 0.22 μm film thickness; Teknokroma, Spain).

### Statistical analysis

All data are presented as mean ± standard deviation. The assumption of normality was verified using the Shapiro-Wilk test. For comparisons of growth and survival metrics across the multiple density treatments within each dietary group, a one-way analysis of variance (ANOVA) was applied to normally distributed data. Where ANOVA identified significant differences, post-hoc comparisons were conducted using Duncan's multiple range test at a 95% confidence level. To evaluate the effect of diet at each stocking density, independent-samples t-tests were conducted comparing commercial and waste feeds. A p-value < 0.05 was considered statistically significant. All analyses were performed using SPSS software (Version 20, IBM Corp., Armonk, NY, USA).

## Results

### Comparative proximate analysis of feed sources

The nutritional composition of the two feed sources differed significantly ($P < 0.05$), establishing a clear baseline for the study (Table 1). The commercial sturgeon feed was nutritionally superior, containing significantly higher crude protein (46.14%) and crude fat content compared to the sturgeon byproducts (28.61% and 13.21%, respectively). Moisture content also differed significantly between the feeds ($P < 0.05$), while ash content was not significantly different ($P > 0.05$). The remainder of the feed composition (46.49% in beluga sturgeon by-products and 27.79% in formulated beluga feed) was not measured directly; this fraction most likely comprises nitrogen-free extract (readily available carbohydrates such as starches and sugar), crude fibre (insoluble carbohydrates) and other non-protein organic constituents. Differences in carbohydrate/fibre content can affect feed digestibility and energy availability and therefore may have contributed to the observed differences in polychaete performance.

### Growth performance and survival of *H. diversicolor*

Stocking density exerted a significant influence ($P < 0.05$) on most growth and survival parameters of *H. diversicolor* (Table 2). A clear inverse relationship was observed between density and individual growth performance. Worms cultured at the lowest density (C1: 1000 ind./m²) yielded the highest final individual weight (0.44 g), specific growth rate

**Table 1. Proximate composition (%; mean ± SD) of beluga sturgeon (*Huso huso*) formulated feed and processing by-products used as experimental diets. Superscript letters indicate statistically significant differences between feed types (P < 0.05).**

| Parameters | Beluga sturgeon byproducts | Beluga sturgeon feed |
|---|---|---|
| Moisture (%) | 7.84 ± 0.13 | 8.12 ± 0.27[A] |
| Crude protein (%) | 28.61 ± 0.06 | 46.14 ± 0.12[A] |
| Crude fat (%) | 13.21 ± 0.19 | 14.68 ± 0.13[A] |
| Ash (%) | 3.85 ± 0.04 | 3.27 ± 0.11 |

The remaining fraction (100 − moisture − crude protein − crude fat − ash) likely represents nitrogen-free extract (carbohydrates), crude fibre, and other minor organic constituents that were not quantified in the present study.

**Table 2. Growth and survival parameters of *Hediste diversicolor* under commercial (C) and sturgeon waste (W) feeding regimes across different stocking densities. For statistical significance, lowercase letters indicate differences among commercial feed groups (C1–C4), and uppercase letters indicate differences among waste feed groups (W1–W4) (P <0.05).**

| Parameters | C1 | C2 | C3 | C4 | W1 | W2 | W3 | W4 |
|---|---|---|---|---|---|---|---|---|
| Initial mean weight (g) | 0.01±0.00 | 0.01±0.00 | 0.01±0.00 | 0.02±0.00 | 0.02±0.00 | 0.02±0.00 | 0.02±0.00 | 0.02±0.00 |
| Final mean weight (g) | 0.44±0.02[d] | 0.38±0.02[c] | 0.27±0.01[b] | 0.12±0.01[a] | 0.12±0.00[C] | 0.10±0.00[B] | 0.08±0.00[A] | 0.07±0.00[A] |
| Initial biomass (g/m²) | 17.32±2.17[a] | 38.69±7.99[b] | 62.15±12.69[c] | 92.99±8.51[d] | 17.01±1.75[A] | 42.56±5.23[B] | 65.41±14.98[C] | 86.97±9.08[D] |
| Final biomass (g/m²) | 392.94±20.34[a] | 749.98±68.16[c] | 897.35±46.85[d] | 522.50±40.11[b] | 98.17±3.62[A] | 206.93±12.31[B] | 261.32±33.59[C] | 283.98±44.16[C] |
| Biomass increase (g/m²) | 375.16±18.53[a] | 711.28±60.58[c] | 835.20±38.46[d] | 429.51±45.73[b] | 81.16±5.11[A] | 164.36±12.15[B] | 195.91±21.32[B] | 197.01±37.60[B] |
| Initial density (ind/m²) | 1000±0.0 | 2500±0.0 | 5000±0.0 | 7500±0.0 | 1000±0.0 | 2500±0.0 | 5000±0.0 | 7500±0.0 |
| Final density (ind/m²) | 886.66±42.52[a] | 1950.00±260.57[b] | 3300.00±371.61[c] | 4325.00±377.19[d] | 803.33±20.20[A] | 1948.33±120.03[B] | 3233.33±470.56[C] | 3956.66±691.23[C] |
| Survival rate (%) | 88.66±4.25[c] | 81.66±4.11[c] | 67.66±5.34[b] | 57.66±5.02[a] | 80.33±2.02[C] | 77.93±1.80[BC] | 64.66±9.41[AB] | 52.75±9.21[A] |
| Specific growth rate (%/day) | 3.47±0.10[c] | 3.30±0.13[c] | 2.98±0.18[b] | 1.91±0.16[a] | 1.95±0.15[C] | 1.76±0.14[BC] | 1.55±0.13[AB] | 1.30±0.10[A] |
| Food consumed (g) | 229.66±5.13 [a] | 573.33±6.65 [b] | 1025.33±4.04 [c] | 1440.33±8.02 [d] | 230.66±3.05 [A] | 571.66±3.51 [B] | 1026.33±8.18 [C] | 1441.33±3.51 [D] |
| FCR | 0.79±0.06 [a] | 0.8±0.06 [a] | 1.22±0.06 [b] | 3.38±0.37 [c] | 2.84±0.19 [A] | 3.49±0.28 [A] | 5.28±0.64 [B] | 7.49±1.39 [C] |
| Daily growth (g/day) | 0.004±0.00[c] | 0.004±0.00[d] | 0.002±0.00[b] | 0.001±0.00[a] | 0.001±0.00[C] | 0.0009±0.00[B] | 0.0007±0.00[A] | 0.0006±0.00[A] |

(SGR: 3.47%/day), and daily growth rate (DGR: 0.004 g/day), with all values being significantly greater ($P < 0.05$) than those in higher density treatments. In contrast, biomass production per unit area was maximized at an intermediate density. The C3 treatment (5000 ind./m²) produced the highest final biomass (897.35 g/m²) and biomass increase (835.20 g/m²), which were significantly greater ($P < 0.05$) than all other treatments. Survival rate was numerically highest at the lowest density (88.66%) but was not statistically different from that at 2500 ind./m² (81.66%); both of these groups showed significantly higher survival than the higher density treatments. The highest density group (C4: 7500 ind./m²) resulted in the poorest individual growth metrics and the lowest survival rate (57.66%), though its final biomass remained significantly higher than that of the lowest density group. These results identify C3 (5000 ind./m²) as the optimal density for maximizing biomass yield of *H. diversicolor* fed a commercial diet.

Stocking density significantly affected the growth and survival of *H. diversicolor* fed sturgeon waste ($P < 0.05$; Table 2). Individual growth performance exhibited a strong inverse relationship with density. Worms at the lowest density (W1: 1000 ind./m²) achieved the highest final individual weight (0.12 g), specific growth rate (SGR: 1.95%/day), and daily growth rate (0.001 g/day), with values significantly greater than those in higher density treatments. In contrast, biomass production per unit area increased with stocking density. The highest final biomass was achieved at the highest densities, with no significant difference observed between W3 (5000 ind./m²; 261.32 g/m²) and W4 (7500 ind./m²; 283.98 g/m²); both were significantly greater than biomass at lower densities. Similarly, biomass increase did not differ significantly among the W2, W3, and W4 treatments. This increase in total biomass output came at the cost of individual welfare, as survival rate significantly declined with increasing density, from 80.33% at 1000 ind./m² to 52.75% at 7500 ind./m². These results suggest that for biomass production on a waste diet, a density of 5000 ind./m² represents a more optimal balance, yielding high biomass output comparable to 7500 ind./m² but with significantly superior survival rates.

Diet and stocking density significantly influenced the production performance of *H. diversicolor* (Fig 2). A significant interaction was observed for biomass increase. For the C group, biomass production peaked at an intermediate density (C3). In contrast, biomass on the byproduct diet (W) increased linearly with density, resulting in a significantly higher yield ($P < 0.05$) at the group W4 compared to its commercial feed counterpart (C4) (Fig 2A). However, the SGR was significantly higher for the commercial feed at all corresponding densities ($P < 0.05$; Fig 2B). While survival rates decreased with increasing density for both diets, there was no significant difference in survival between the two feed types at any density ($P > 0.05$). This indicates that the byproduct diet can support higher total biomass at high density without compromising survival, though it results in significantly lower individual growth rates (Fig 2C).

## Proximate composition of *H. diversicolor*

Stocking density significantly altered the body composition of *H. diversicolor* ($P < 0.05$; Table 3). Stocking density significantly influenced the body composition of worms fed the commercial diet (C) ($P < 0.05$; Table 3). Crude protein content was significantly higher in all commercial groups (C1 to C4) compared with the corresponding waste diet groups at the same density (independent-samples t-tests, $P < 0.05$). Crude fat was significantly higher in C3 compared with W3 ($P < 0.05$), while other fat comparisons between diets were not significant. Ash content was significantly higher in C1 and C3 compared with W1 and W3, respectively ($P < 0.05$). Moisture content differed significantly between diets only at W1, W2, and W4 ($P < 0.05$), and remained largely stable across the other densities. Group C2 had a significantly higher crude protein content than all other groups ($P < 0.05$). Crude fat was significantly higher in C1 and lowest in C4 ($P < 0.05$). Ash content showed the most dramatic response, being significantly higher in group C1 and lower in group C4. Moisture content remained stable and was not significantly affected ($P > 0.05$).

Stocking density significantly influenced the body composition of *H. diversicolor* fed sturgeon waste ($P < 0.05$; Table 3). Group W2 had a significantly higher crude protein content than all other groups ($P < 0.05$). Crude fat was significantly highest in the group W1 ($P < 0.05$). Ash content showed a distinct pattern, being significantly higher in the group W1 and W4 ($P < 0.05$). Moisture content was stable and not significantly affected by density ($P > 0.05$).

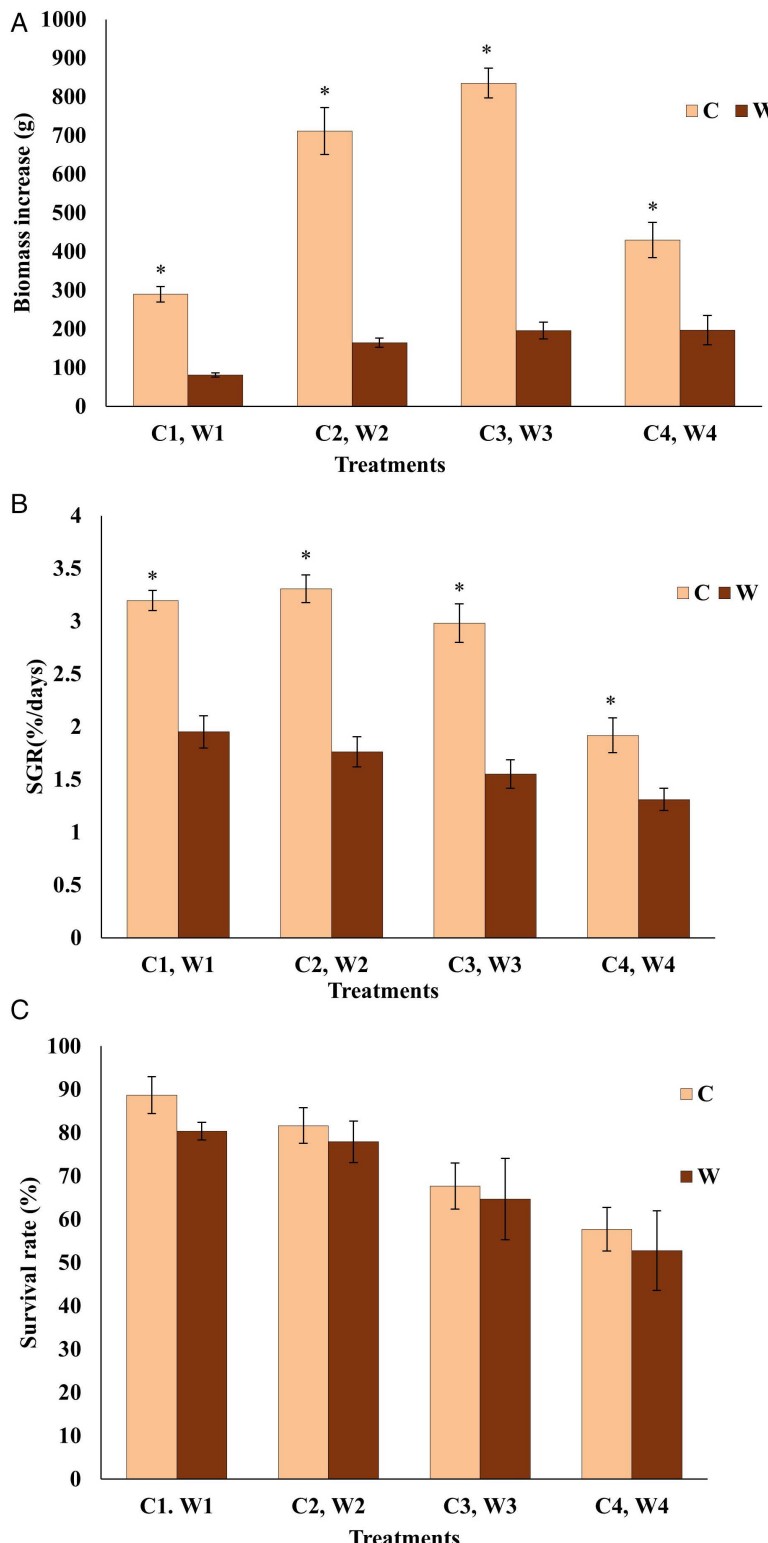

**Fig 2. Comparison of biomass increase (A), specific growth rate (B), and survival rate (C) in *H. diversicolor* cultured at different stocking densities and fed either beluga sturgeon (*Huso huso*) formulated feed (C) or processing byproducts (W).** Asterisks (*) indicate significant differences between feed types at each density (independent samples t-tests, $P < 0.05$).

**Table 3. Proximate composition (%; mean±SD) of *Hediste diversicolor* under commercial (C) and waste (W) feeding regimes across four stocking densities (60 d). Lowercase letters indicate significant differences among commercial feed groups (C1–C4), and uppercase letters indicate significant differences among waste feed groups (W1–W4) (one-way ANOVA, Duncan's test, P<0.05). Asterisks (\*) indicate significant differences between diets within the same density (independent-samples t-test, P<0.05).**

| Parameters | C1 | C2 | C3 | C4 | W1 | W2 | W3 | W4 |
|---|---|---|---|---|---|---|---|---|
| Moisture (%) | 85.23±0.06 | 85.34±0.07* | 85.18±0.08 | 85.26±0.13 | 85.28±0.05* | 85.12±0.08 | 85.24±0.03 | 85.62±0.04* |
| Crude protein (%) | 8.36±0.09[b]* | 9.23±0.16[c]* | 7.74±0.05[a]* | 8.12±0.17[b]* | 7.41±0.27[A] | 8.62±0.32[B] | 7.28±0.11[A] | 7.31±0.02[A] |
| Crude fat (%) | 4.34±0.27[b] | 3.82±0.29[ab] | 4.19±0.31[ab]* | 3.42±0.38[a] | 3.83±0.19[A] | 3.29±0.21[B] | 3.18±0.25[A] | 3.25±0.14[A] |
| Ash (%) | 2.78±0.08[c]* | 1.39±0.04[a] | 1.75±0.14[b]* | 1.28±0.11[a] | 1.18±0.12[B] | 1.34±0.25[A] | 1.21±0.14[A] | 1.67±0.16[B]* |

### Fatty acid profile of *H. diversicolor*

Stocking density significantly altered the fatty acid composition of *H. diversicolor* fed a commercial diet (*P*<0.05; Table 4). This demonstrates that the intermediate stocking density (2500 ind./m², C2) is optimal for enhancing the nutritional value of worms as a live feed resource, particularly by enriching their content of beneficial omega-3 fatty acids. The nutritional profile was optimized at the intermediate density of 2500 ind./m² (C2). This treatment yielded the most favorable nutritional profile, characterized by the highest total PUFA content (46.85%), the most beneficial n-3/n-6 ratio (0.39), and the lowest SFA level (18.57%). This improvement was driven primarily by a dramatically higher concentration of α-linolenic acid (C18:3n-3; 8.41±0.29%) compared to other densities. Arachidonic acid (ARA, C20:4n-6) in C2 was 0.34±0.07% (significantly lower than C1: 0.48±0.02% and C4: 0.51±0.08%), whereas eicosapentaenoic acid (EPA, C20:5n-3) was maintained at relatively high levels in C1 and C2 (C1: 2.88±0.13%; C2: 2.74±0.27%), both significantly higher than in C3 (2.37±0.32%) and C4 (2.40±0.21%). Docosahexaenoic acid (DHA, C22:6n-3) peaked in C1 (1.51±0.18%), was intermediate in C4 (1.12±0.09%) and C2 (1.06±0.04%), and was lowest at C3 (0.97±0.07%). These data indicate that C2 provides the best overall balance of PUFA, a superior n-3/n-6 ratio, and low SFA (see Table 4 for full fatty-acid profiles and statistical comparisons).

Stocking density significantly altered the fatty acid profile of *H. diversicolor* fed sturgeon byproducts (*P*<0.05; Table 4). The most nutritionally balanced profile was observed at the intermediate density of 2500 ind./m² (W2), which yielded the most favorable n-3/n-6 ratio (0.22) and the highest total n-3 fatty acid content (5.69%). In contrast, the high density of 5000 ind./m² (W3) resulted in a significantly poorer nutritional profile, characterized by the highest saturated fatty acid (SFA) content (42.33%), the lowest polyunsaturated fatty acid (PUFA) content (25.38%), and a consequently high SFA/Unsaturated ratio (0.73).

Levels of key long-chain PUFAs were also significantly affected. Eicosapentaenoic acid (EPA) was highest at the lowest density (W1: 1.57%), while docosahexaenoic acid (DHA) was highest and statistically similar in W2 and W3 (1.63–1.69%). The DHA/EPA ratio, an indicator of metabolic conversion, increased significantly with density, reaching its maximum in the W3 treatment (1.87). The highest density (W4: 7500 ind./m²) was associated with the lowest overall n-3 content (2.85%). These results demonstrate that stocking density is a critical factor modulating the nutritional quality of waste-fed polychaetes, with an intermediate density (2500 ind./m²) providing the optimal balance for producing a high-quality feed ingredient.

Independent-samples t-tests comparing the two diets at each stocking density (C vs W within density) revealed consistent diet-dependent differences in fatty-acid composition (Table 4). At the lowest density (C1 vs W1), worms fed the commercial diet had significantly higher total PUFA (37.92±3.25% vs 33.33±2.25%, P<0.05), higher EPA (2.88±0.13% vs 1.57±0.21%, P<0.05) and higher DHA (1.51±0.18% vs 1.07±0.08%, P<0.05), whereas waste-fed worms exhibited a significantly higher Σ Saturated and Sat/Unsat ratio (29.90±2.32% and 0.42±0.03, respectively; both P<0.05). At the intermediate commercial density (C2 vs W2), commercial-fed worms were significantly enriched in total PUFA (46.85±2.39% vs 34.66±2.17%, P<0.05), Σn-3 (12.22±0.93% vs 5.69±0.92%, P<0.05) and EPA (2.74±0.27% vs 1.27±0.18%, P<0.05), while the waste diet produced higher ΣSaturated (31.24±3.61%, P<0.05) and a higher DHA level in T2-2 (1.63±0.27%,

**Table 4. Fatty acid profiles of *Hediste diversicolor* polychaetes fed commercial feed (C) and beluga sturgeon (*Huso huso*) by-product feed (W) at four stocking densities over a 60-day culture period.** Lowercase letters indicate significant differences among stocking densities within the commercial feed groups (C1–C4), and uppercase letters indicate significant differences among stocking densities within the waste feed groups (W1–W4), as determined by one-way ANOVA followed by Duncan's multiple range test (P<0.05). Asterisks (*) indicate significant differences between diets at the same stocking density based on independent-samples t-tests (P<0.05).

| Parameters | C1 | C2 | C3 | C4 | W1 | W2 | W3 | W4 |
|---|---|---|---|---|---|---|---|---|
| **Saturates (SFA)** | | | | | | | | |
| C12:0 | 1.88±0.02c | 0.81±0.06b* | 0.41±0.06a | 2.41±0.08d* | 2.6±0.28D* | 0.51±0.14A | 2.25±0.23C* | 1.33±0.31B |
| C14:0 | 0.83±0.15b | 0.63±0.05a | 1.03±0.12c | 1.12±0.08c | 3.22±0.37C* | 1.11±0.22A* | 4.30±0.63D* | 2.09±0.51B* |
| C16:0 | 11.75±0.34a | 11.95±1.44a | 15.16±1.31b | 11.39±1.92a | 18.95±1.23D* | 20.66±2.14A* | 26.72±1.59B* | 20.03±1.32A* |
| C18:0 | 2.47±0.13a | 2.59±0.25a | 2.82±0.36a | 2.85±0.97a | 4.10±0.28B* | 2.75±0.13A | 6.78±0.38C* | 4.95±0.87B* |
| C20:0 | 0.60±0.04b | 0.52±0.07ab | 0.42±0.02a | 1.30±0.05c | 1.02±0.05A* | 1.32±0.07B* | 1.9±0.03C* | 1.2±0.06AB |
| C22:0 | 4.66±0.17b* | 2.05±0.04a | 2.20±0.08a | 4.58±0.85b* | 1.51±0.62A | 4.88±0.37C* | 2.6±0.26B* | 2.53±0.42B |
| **Monounsaturated (MUFA)** | | | | | | | | |
| C16:1 | 3.85±0.61c* | 1.96±0.19a | 3.11±0.26b* | 3.78±0.34c* | 2.22±0.19A | 2.48±0.25A* | 2.12±0.17A | 2.02±0.12A |
| C18:1n-9 | 32.17±1.83ab* | 30.52±2.61a | 37.10±1.34b | 28.98±2.66a | 30.8±2.61B | 29.43±3.66AB | 25.5±2.62A | 27.57±2.56AB |
| C20:1 | 2.34±0.23c* | 1.12±0.27a | 2.04±0.61b* | 2.46±0.18c | 1.09±0.06A | 1.20±0.03A | 1.25±0.04A | 2.09±0.08B |
| C22:1n-9 | 1.47±0.08b | 0.95±0.05a | 1.04±0.06a | 1.58±0.03a | 1.11±0.06AB | 0.95±0.05A | 1.10±0.02A | 1.39±0.09B |
| **Polyunsaturated (PUFA)** | | | | | | | | |
| C18:2n-6 | 27.65±1.58b | 30.26±2.65c* | 24.60±1.67a | 29.51±1.69bc | 26.04±1.32B | 24.64±1.61B | 20.33±1.59A | 27.89±2.11A |
| C18:3n-3 | 1.64±0.16a | 8.41±0.29c* | 3.69±0.36b* | 2.15±0.28a* | 1.84±0.05C | 2.78±0.04D | 0.57±0.06A | 1.01±0.02B |
| C20:2 | 3.73±0.10b* | 3.98±0.24b | 2.61±0.64a* | 3.78±0.32b | 2.39±0.17B | 4.08±0.24D | 1.64±0.09A | 3.75±0.37C |
| C20:4n-6 | 0.48±0.02b | 0.34±0.07a* | 0.35±0.03a* | 0.51±0.08b* | 0.39±0.03B | 0.22±0.05A | 0.23±0.06A | 0.24±0.04A |
| C20:5n-3 (EPA) | 2.88±0.13b* | 2.74±0.27b* | 2.37±0.32a* | 2.40±0.21a* | 1.57±0.21C | 1.27±0.18B | 0.90±0.05A | 0.76±0.02D |
| C22:6n-3 (DHA) | 1.51±0.18c* | 1.06±0.04ab | 0.97±0.07a | 1.12±0.09b | 1.07±0.08A | 1.63±0.27B* | 1.69±0.31B* | 1.07±0.06A |
| **Profile variables** | | | | | | | | |
| % Σ Saturated | 22.21±1.69b | 18.57±1.13a | 22.07±1.82b | 23.68±1.39b | 29.90±2.32A* | 31.24±3.61A* | 42.33±2.41B* | 32.15±1.83A* |
| % Σ MUFA | 39.85±2.34c | 34.57±1.28a | 43.31±2.37d* | 36.81±1.52b | 36.75±3.66B | 34.08±2.52AB | 32.28±2.61A | 33.10±2.41AB |
| % Σ PUFA | 37.92±3.25b* | 46.85±2.39d* | 34.61±3.62a* | 39.49±2.20c* | 33.33±2.25B | 34.66±2.17B | 25.38±1.52A | 34.74±2.16B |
| % Σ Unsat | 77.78±2.56a* | 81.42±1.67b* | 77.92±2.89a* | 76.31±3.28a* | 70.09±3.62B | 68.75±2.51B | 57.66±2.66A | 67.84±1.59B |
| Sat/Unsat ratio | 0.28±0.04a | 0.22±0.06a | 0.28±0.08a | 0.31±0.07a | 0.42±0.03A* | 0.45±0.06A* | 0.73±0.07B* | 0.47±0.04A* |
| % Σn-3 in FA | 6.05±1.61a | 12.22±0.93c* | 7.04±0.57b* | 5.68±0.27a* | 4.50±0.25B | 5.69±0.92C | 3.17±0.53A | 2.85±0.71A |
| % Σn-6 in FA | 28.13±2.35b | 30.64±2.58b* | 24.95±1.83a | 30.02±2.51b | 26.43±1.91B | 24.87±1.65B | 20.56±1.32A | 28.13±2.14C |
| Σ n-3/ Σ n-6 | 0.21±0.03a | 0.39±0.08c* | 0.28±0.06b* | 0.18±0.02a* | 0.17±0.02B | 0.22±0.03C | 0.15±0.01B | 0.10±0.02A |
| DHA/EPA | 0.52±0.07b | 0.38±0.06a | 0.40±0.09a | 0.46±0.05b | 0.67±0.04A* | 1.28±0.08B* | 1.87±0.09C* | 1.39±0.06B* |

P<0.05). At the higher intermediate density (C3 vs W3), commercial-fed worms again had significantly higher EPA and several PUFA measures (e.g., ΣPUFA 34.61±3.62% vs 25.38±1.52%, P<0.05), whereas waste-fed worms showed substantially higher ΣSFA and Sat/Unsat ratio (42.33±2.41% and 0.73±0.07, respectively; P<0.05). Finally, at the highest density (C4 vs W4), commercial-fed worms retained higher ΣPUFA and Σn-3 (39.49±2.20% and 5.68±0.27%, respectively) and higher EPA (2.40±0.21% vs 0.76±0.02%; P<0.05), whereas waste-fed worms had higher ΣSFA and a higher DHA/EPA ratio (both P<0.05). The full set of per-fatty-acid t-test results is presented in Table 4 (asterisks indicate P<0.05).

### Fatty acid profile of beluga sturgeon feed and processing byproducts

A comparative analysis revealed significant differences in the fatty acid profiles of the original sturgeon feed and the resulting processing byproducts (*P*<0.05; Table 5). The byproducts were characterized by significantly higher total

**Table 5. Fatty acid profile analysis of beluga sturgeon (*Huso huso*) formulated feed and processing byproducts. Values are expressed as mean percentage of total fatty acids ± SD (n = 4). Superscript letters indicate statistically significant differences between feed types (P < 0.05).**

| Parameters | Beluga sturgeon byproducts | Beluga sturgeon feed |
|---|---|---|
| **Saturates (SFA)** | | |
| C12:0 | 0.42 ± 0.12 | 0.96 ± 0.07[A] |
| C14:0 | 0.57 ± 0.17 | 1.01 ± 0.09[A] |
| C16:0 | 31.61 ± 1.54[A] | 19.38 ± 1.36 |
| C18:0 | 3.87 ± 0.61 | 9.11 ± 0.33[A] |
| C20:0 | 2.33 ± 0.09[A] | 1.36 ± 0.09 |
| C22:0 | 0.72 ± 0.13[A] | 0.51 ± 0.51 |
| **Monounsaturated (MUFA)** | | |
| C16:1 | 3.77 ± 0.24[A] | 2.14 ± 0.29 |
| C18:1n-9 | 27.60 ± 3.59 | 31.32 ± 1.87[A] |
| C20:1 | 1.47 ± 0.07[A] | 0.31 ± 0.29 |
| C22:1n-9 | 0.36 ± 0.05 | 0.49 ± 0.07[A] |
| **Polyunsaturated (PUFA)** | | |
| C18:2n-6 | 22.91 ± 2.28 | 28.03 ± 1.22[A] |
| C18:3n-3 | 2.13 ± 0.09 | 2.72 ± 0.13[A] |
| C20:2 | 0.53 ± 0.04 | 0.54 ± 0.81 |
| C20:4n-6 | 0.15 ± 0.06 | 0.23 ± 0.06[A] |
| C20:5n-3 (EPA) | 0.53 ± 0.07 | 0.85 ± 0.11[A] |
| C22:6n-3 (DHA) | 0.45 ± 0.8 | 0.57 ± 0.05[A] |
| **Profile variables** | | |
| % Σ Saturated | 39.54 ± 1.56[A] | 32.91 ± 1.24 |
| % Σ MUFA | 33.21 ± 2.67 | 35.03 ± 1.65[A] |
| % Σ PUFA | 26.71 ± 2.57 | 33.06 ± 1.85[A] |
| % Σ Unsat | 59.93 ± 3.91 | 68.09 ± 2.57[A] |
| Sat/Unsat ratio | 0.65 ± 0.06[A] | 0.48 ± 0.09 |
| % Σn-3 in FA | 3.11 ± 0.31 | 4.14 ± 0.85[A] |
| % Σn-6 in FA | 23.06 ± 1.56 | 28.37 ± 1.28[A] |
| Σ n-3/ Σ n-6 | 0.13 ± 0.01 | 0.14 ± 0.01 |
| DHA/EPA | 0.84 ± 0.05[A] | 0.66 ± 0.04 |

saturated fatty acid (SFA) content (39.54% vs. 32.91%) and a consequently less favourable saturation ratio (0.65 vs. 0.48) compared to the commercial feed. Conversely, the original feed had a significantly higher total polyunsaturated fatty acid (PUFA) content (33.06% vs. 26.71%), which was primarily driven by a higher n-6 fatty acid content. Most critically, the commercial feed contained significantly higher levels of the essential long-chain omega-3 fatty acids, eicosapentaenoic acid (EPA; 0.85% vs. 0.53%) and docosahexaenoic acid (DHA; 0.57% vs. 0.45%). Despite the absolute levels of EPA and DHA being lower in the byproducts, their DHA/EPA ratio was significantly higher (0.84 vs. 0.66). No significant difference was observed in the total monounsaturated fatty acid (MUFA) or the overall n-3/n-6 ratio between the two dietary sources.

## Discussion

The present study demonstrates that integrating *H. diversicolor* into beluga sturgeon aquaculture operations effectively transform waste into valuable biomass, aligning with the principles of a circular economy. This system addresses the

critical need for sustainable organic waste management in recirculating aquaculture systems while producing a nutritionally rich polychaete suitable as a live feed supplement. Our findings reveal that solid waste particulates from sturgeon culture retain significant nutritional value, containing 7.28–8.62% crude protein and 3.18–3.83% lipids on a wet weight basis, making them a viable substrate for *H. diversicolor* rearing systems [39–41].

A key finding of this study is the pronounced trade-off between individual growth rate and total biomass yield, which is critically influenced by both stocking density and diet type. The observed inverse relationship between stocking density and individual growth performance, where the lowest density (1000 ind./m²) yielded the highest SGR is a well-documented phenomenon in polychaete culture [42–44]. This pattern is primarily attributed to intensified intraspecific competition for space and food resources at elevated densities [41–44]. Our results aligned with studies on *H. diversicolor* reporting reduced growth rates at higher densities due to behavioral changes and increased energy expenditure on competition rather than biomass accumulation [41]. Notably, survival rates remained statistically comparable between byproduct-fed (52.75–80.33%) and formulated feed groups (57.66–88.66%), though observed cannibalistic behavior in high-density waste treatments [45–48], may explain marginal survival reductions.

The diet-dependent biomass optimization was particularly noteworthy. For the commercial feed, biomass peaked at an intermediate density of 5000 ind./m², following a typical parabolic response where increased competition eventually outweighs the benefits of higher stocking numbers. In contrast, for the sturgeon waste diet, biomass production showed a positive linear relationship with density, culminating in the highest yield at 7500 ind./m². This suggests that for nutritionally inferior waste diets, maximizing individual numbers becomes a viable strategy to compensate for slower individual growth rates and achieve high total system productivity. This aligns with the economic imperative in aquaculture to intensify production, though it must be balanced against animal welfare concerns [46].

The proximate and fatty acid composition of *H. diversicolor* was significantly influenced by the diet, with stocking density playing a secondary modulating role. The significantly higher crude protein content observed in polychaetes fed the commercial diet is supported by the direct comparisons at each stocking density (Table 3), where all commercial groups (C1 to C4) exhibited significantly higher protein than the corresponding waste-fed groups (W) (independent-samples t-tests, $P < 0.05$). Similarly, crude fat was significantly higher in C3 compared with W3, and ash content was significantly higher in C1 and C3 compared with W1 and W3, respectively, while moisture content remained largely stable across diets. These differences reflect the superior nutritional profile of the formulated commercial feed, which contained higher protein and lipid content than the sturgeon byproduct feed [46]. This finding underscores a fundamental principle in aquaculture: the proximate composition and overall nutritional quality of cultured organisms are strongly influenced by the composition of the feed provided [47].In addition to protein and lipid content, differences in the unmeasured fraction of the two feeds, likely comprising carbohydrates, fibre, and other nitrogen-free extractives, may have influenced digestibility and energy availability, thereby contributing to the observed differences in polychaete growth and biomass production.

Notably, waste-fed worms maintained nutritionally significant levels of essential long-chain PUFAs, eicosapentaenoic acid (EPA) and docosahexaenoic acid (DHA), despite the poorer dietary input. The elevated PUFA content in worm tissues compared to the ingested waste provides strong evidence for the *de novo* biosynthetic capability of *H. diversicolor* to elongate and desaturate dietary fatty acids [46–48]. This metabolic capability enhances their value in nutrient recycling within IMTA systems, as they can effectively upgrade nutrients from waste streams into valuable biomolecules [49].

The present results demonstrate that stocking density is a key factor regulating fatty acid metabolism and nutritional quality in waste-fed *H. diversicolor*. In polychaetes fed sturgeon byproducts, increasing density markedly altered lipid composition, with the intermediate density (2500 ind./m²; T2-2) producing the most nutritionally favorable profile, characterized by a relatively high n-3 fatty acid content and a balanced n-3/n-6 ratio. In contrast, higher densities led to a deterioration of fatty acid quality, as evidenced by the progressive increase in saturated fatty acids (SFA), elevated SFA/Unsaturated ratios, and a decline in total n-3 fatty acids, reaching a minimum at the highest density (W4; 7500 ind./m²). Similar density-dependent shifts in lipid profiles have been reported in polychaetes and other benthic invertebrates,

where crowding stress reduces feeding efficiency and alters lipid biosynthesis and deposition pathways [50,51]. The increase in the DHA/EPA ratio with increasing density, peaking at W3, suggests density-related modulation of long-chain PUFA metabolism. Elevated DHA/EPA ratios under stressful or nutritionally constrained conditions have been associated with selective retention of DHA and reduced EPA availability, reflecting metabolic prioritization rather than dietary input alone [52,53]. The sharp decline in total n-3 fatty acids at the highest density further indicates that excessive crowding compromises the nutritional value of waste-fed polychaetes, likely through reduced ingestion rates and altered assimilation efficiency.

Direct comparisons between diets at the same stocking density confirmed that diet quality consistently overrode density effects in determining fatty acid composition. Independent-samples t-tests revealed that polychaetes fed the commercial diet exhibited significantly higher total PUFA, EPA, and n-3 fatty acids across all densities compared with waste-fed worms, while the latter consistently showed higher SFA content and SFA/Unsaturated ratios. These findings are consistent with previous studies demonstrating that polychaetes largely reflect the fatty acid signature of their diet, with limited capacity to biosynthesize long-chain n-3 PUFAs when dietary precursors are scarce [54–56]. The superior PUFA and EPA levels in commercial-fed worms therefore reflect the higher nutritional quality and lipid composition of the formulated feed relative to sturgeon processing byproducts.

Taken together, these results highlight a strong interaction between stocking density and dietary source in shaping the fatty acid profile of *H. diversicolor*. While waste-based feeds can support acceptable fatty acid profiles at moderate densities, excessive stocking density substantially reduces their nutritional value. From an applied perspective, maintaining intermediate densities (≈2500 ind./m²) appears critical for optimizing the fatty acid quality of waste-fed polychaetes intended for use as a sustainable aquafeed ingredient. The current study system demonstrated high efficiency in converting waste into valuable biomass, validating its potential for sustainable aquaculture. The production of up to 283.98 g/m² of worm biomass from waste at the highest density is a key indicator of the system's bioremediation efficacy. This efficient conversion of waste into animal biomass is a cornerstone of IMTA and aligns with research promoting the use of polychaetes for biomitigation in intensive aquaculture effluents [19,40,41,45,57,58].

The study showed notable nutrient retention efficiencies, particularly for essential FAs like EPA, with 89−127% recovery efficiency from waste streams. This ability to conserve and upgrade nutrients from waste streams is a critical function that reduces the environmental footprint of aquaculture operations while creating a valuable co-product [49–55]. The performance of the system is comparable and, in some respects, similar to other bioremediation strategies, including the organic matter conversion rates reported for *H. diversicolor* in salmon aquaculture systems [47,59]. Although carbohydrate fractions were not quantified directly in the present study, their known contribution to dietary energy and digestibility suggests that future work incorporating fibre and nitrogen-free extract analyses would further clarify feed–performance relationships in polychaetes culture systems.

## Conclusion

The study identifies the worm *H. diversicolor* as an excellent candidate for sustainable aquaculture. It can effectively convert fish waste into nutritious biomass, promoting a more circular system. The research offers a scalable model centred on density optimization and dietary enhancement. For premium markets requiring large, high-quality worms (e.g., bait), a lower density of 1000–2500 ind./m² is recommended. For the primary goals of waste treatment and producing bulk biomass, a higher density of 5000–7500 ind./m² is optimal. The nutritional profile, specifically PUFA content, can be enhanced through strategic lipid supplementation from microalgae or oils without compromising the economic benefits of using waste. This model provides a scalable solution to reduce aquaculture's environmental impact and create economic value from waste. Future research should focus on maximizing nutrient retention through seasonal cultivation cycles, microbial priming to enhance waste digestibility, and economic modelling of optimal worm-to-fish ratios, contributing to the sustainable development of aquaculture globally.

## Supporting information

**S1 Dataset. Complete dataset.** This file contains all data generated and analysed during this study.
(XLSX)

## Acknowledgments

The authors would also like to thank the International Sturgeon Research Institute (ISRI, Guilan province, Iran) for technical assistance throughout the project.

## Author contributions

**Conceptualization:** Zabihollah Pajand, Ayoub Yousefi, Arash Lebria, Younes Golalipour, Parham Pajand, Mohebali Pourgholam.

**Data curation:** Zabihollah Pajand, Ayoub Yousefi, Hamed Abdollahpour.

**Formal analysis:** Zabihollah Pajand, Ayoub Yousefi, Esmaeil Hosseinnia, Arash Lebria, Younes Golalipour, Parham Pajand, Mohebali Pourgholam.

**Funding acquisition:** Zabihollah Pajand.

**Investigation:** Zabihollah Pajand.

**Methodology:** Zabihollah Pajand, Ayoub Yousefi, Esmaeil Hosseinnia, Arash Lebria.

**Project administration:** Zabihollah Pajand.

**Resources:** Zabihollah Pajand, Ayoub Yousefi, Esmaeil Hosseinnia, Parham Pajand.

**Software:** Zabihollah Pajand.

**Supervision:** Zabihollah Pajand.

**Validation:** Zabihollah Pajand.

**Visualization:** Zabihollah Pajand.

**Writing – original draft:** Zabihollah Pajand, Hamed Abdollahpour, Naghmeh Jafari Pastaki.

**Writing – review & editing:** Zabihollah Pajand, Hamed Abdollahpour, Naghmeh Jafari Pastaki.

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
