## [Decision Letter · Decision Letter 0]

16 Jan 2026

Dear Dr. Pajand,

Thank you for submitting your manuscript to PLOS ONE. After careful consideration, we feel that it has merit but does not fully meet PLOS ONE’s publication criteria as it currently stands. Therefore, we invite you to submit a revised version of the manuscript that addresses the points raised during the review process.

We look forward to receiving your revised manuscript.

Kind regards,

Eman Zahran, Ph.D

Academic Editor

PLOS One

Journal Requirements:\

“This study was financially supported by grants from the Iran National Science Foundation (INSF Contract no. 91002674) program project.”

4. Please note that funding information should not appear in any section or other areas of your manuscript. We will only publish funding information present in the Funding Statement section of the online submission form. Please remove any funding-related text from the manuscript.

5. In the online submission form, you indicated that “The datasets used and/or analysed during the current study are available from the corresponding author on reasonable request.”

Additional Editor Comments:

Dear Authors,

Thank you for submitting your manuscript entitled “Diet and density effects on growth, proximate composition and fatty-acid profile of Hediste diversicolor juveniles” to the journal. The manuscript has now been evaluated by external peer reviewers and assessed at the editorial level.

The reviewers have raised several substantial concerns that must be carefully and comprehensively addressed before the manuscript can be reconsidered. Detailed reviewer comments are provided below to guide you in revising the manuscript.

upon submitting a revised version, please ensure that all reviewer comments are addressed point-by-point in a separate response document.

Thank you for considering the journal for your work. We look forward to receiving your revised submission.

Reviewer's Responses to Questions

**Comments to the Author**

1. Is the manuscript technically sound, and do the data support the conclusions?

Reviewer #1: Yes

Reviewer #2: Yes

2. Has the statistical analysis been performed appropriately and rigorously?

Reviewer #1: Yes

Reviewer #2: No

3. Have the authors made all data underlying the findings in their manuscript fully available?

Reviewer #1: Yes

Reviewer #2: Yes

4. Is the manuscript presented in an intelligible fashion and written in standard English?

Reviewer #1: Yes

Reviewer #2: Yes

Reviewer #1: Introduction

Line 84 to 93 – These two sentence of the this paragraph is unnecessary. Remove it. Instead of this, importance of live prey utilization in both marine and freshwater fish species should have been summarized with using related references (see below). The authors should go for alternative live preys such as polycheates utilization and its importance. Then you can continue fatty acid composition of this polyceates.

Eryalcin, K. (2018). Effects of different commercial feeds and enrichments on biochemical composition and fatty acid profile of rotifer (Brachionus Plicatilis, Muller 1786) and Artemia franciscana. Turkish Journal of Fisheries and Aquatic Sciences, 18.

Turcihan, G., Turgay, E., Yardımcı, R. E., & Eryalçın, K. M. (2021). The effect of feeding with different microalgae on survival, growth, and fatty acid composition of Artemia franciscana metanauplii and on predominant bacterial species of the rearing water. Aquaculture International, 29(5), 2223-2241.

Eryalçın, K. M. (2019). Nutritional value and production performance of the rotifer Brachionus plicatilis Müller, 1786 cultured with different feeds at commercial scale. Aquaculture International, 27(3), 875-890.

Turcihan, G., Isinibilir, M., Zeybek, Y. G., & Eryalçın, K. M. (2022). Effect of different feeds on reproduction performance, nutritional components and fatty acid composition of cladocer water flea (Daphnia magna). Aquaculture Research, 53(6), 2420-2430.

Line 97 – The fatty acid metabolism of H. diversicolor should have been explored. What are the differences comparing rotifer, artemia ? Please see related refences than you can get some clues (see below).

Villena-Rodríguez, A., Monroig, Ó., Hontoria, F., Malzahn, A. M., Hagemann, A., & Navarro, J. C. (2025). Effects of temperature and salinity on the LC-PUFA biosynthesis and composition of the nereid polychaete Hediste diversicolor fed side streams. Aquaculture International, 33(1), 56.

Villena-Rodríguez, A., Navarro, J. C., Varó, I., Aguado-Giménez, F., Pérez-Ara, J., Lizanda, M., ... & Monroig, Ó. (2025). Investigating the potential of dietary iron supplementation to enhance long-chain polyunsaturated fatty acid biosynthesis in Hediste diversicolor. Scientific Reports, 15(1), 42473.

Line 100 – There should be mentioned specifically PUFAs importance in different stage of fish. That means larvae, growth out and broodstock. Each life stage of fish requiere different amount of those fatty acids. The authors should define with related references. This is the important part of study. It should be extended.

Eryalçın, K. M., Ganuza, E., Atalah, E., & Hernández Cruz, M. C. (2015). Nannochloropsis gaditana and Crypthecodinium cohnii, two microalgae as alternative sources of essential fatty acids in early weaning for gilthead seabream. Hidrobiológica, 25(2), 193-202.

Tseng, Y., Eryalçın, K. M., Sivagurunathan, U., Domínguez, D., Hernandez-Cruz, C. M., Boglione, C., ... & Izquierdo, M. (2023). Effects of the dietary supplementation of copper on growth, oxidative stress, fatty acid profile and skeletal development in gilthead seabream (Sparus aurata) larvae. Aquaculture, 568, 739319.

Material and Methods

Line 129 - I advice to put subcaption such as ’’Polyceate Husbandry’’?

Line 164 – The labeling groups should be changed. This can be like Commercial C1, C2, C3, C4 depending on density. And other can be like Waste (W1, W2, W3, W4). In that way, similarity can be removed among treatments. Specification of commercial diets should be given (brand, country, size)

Line 173 to 181 – This paragraph can be removed its because you are already explaining zootechnical measurements (even we don’t call it like that)

Line 202 - How it has been measured and with which kind of tools, brand, country and model should be given.

Results

I think two caption should be get together in one subcaption like ‘’Growth performance and Survival of H. diversicolor’’. Same comment for Proximate composition section. Proximate composition of H.diversicolor should be combined both ‘’…fed commercial feed’’ and ‘’….. fed sturgeon waste’’; combined under one caption.

Line 319 – Proximate and Fatty acid composition of sturgeon waste and commercial diets should be given first.

Line 333 - While mentioning ‘’Fatty acid profiles of H.diversicolor’’ at between Line 326 and 336, ARA, EPA and DHA values should be given.

Line 335 – This paragraph should go up.

Discussion

I don’t know whether PlosOne includes subcaptions or not at Discussion section. Please check !

I think, both tables (Table 1 and Table 2) can be combined.

Same comment for Table 3 and Table 4.

At Table 5, you can use upper superscript letters just like you did in other tables.

Please correct ‘’Sat’’ to ‘’Saturated’’ both fatty acid tables.

At Table 8 – Use upper superscript letters instead of ‘’*’’

Reviewer #2: Both the presentation of results and the statistical analysis require improvement.

Figure 1 – In the figure, it is not clear that each experimental group corresponds to three tanks. This information should be clearly represented. Additionally, the terminology corresponding to the tested densities (i.e. 1, 2, 3 and 4) should be included.

Line 179 – What was the feed-to-worm biomass ratio used in this study?

Proximate composition analysis: How was ash content determined?

Lines 222 and 228 – Total lipids were determined using two different methods (Soxhlet and Folch). Please justify the use of two methodologies and clarify which method corresponds to the results presented.

Line 238 – Please specify the FAME standards used.

Line 240 – Please confirm whether fatty acid results are expressed as mg FA per g of wet tissue and explain the calculation. Tables 6, 7 and 8 present fatty acid data as a percentage of total fatty acids. How do you justify this discrepancy in units?

Line 268 – The identification of the experimental group with a density of 5000 ind./m² (T1–3) must be included.

Lines 298–310 and Tables 3 and 4 –

The sum of moisture, protein, lipids and ash is close to 100%. Does H. diversicolor not contain carbohydrates? Are these results consistent with previous studies?

Additionally, the proximate composition of H. diversicolor obtained with the two different feeds is not compared. The information presented in Tables 3 and 4 should be compiled into a single table. For each tested density, results obtained with the two diets should be directly compared, including appropriate statistical analysis.

Lines 311–324 / Table 5 and Figure 3 –

Information regarding the proximate composition of the two experimental feeds is duplicated. Table 5 and Figure 3 present the same data; therefore, Figure 3 should be removed.

Moreover, the sum of moisture, protein, fat and ash represents only 53.51% and 72.21% of beluga sturgeon by-products and beluga sturgeon feed, respectively. What does the remaining fraction correspond to? Carbohydrates, including fibre? Could differences in these nutrients also help explain differences observed in polychaete performance?

Lines 338–367 –

The fatty acid profiles of H. diversicolor obtained with the two feeds should be compared for each density, including statistical analysis.

Lines 401–405 –

The data supporting these conclusions are not adequately highlighted in the Results section. Please refer to the comments regarding Tables 3 and 4.

**Do you want your identity to be public for this peer review?** For information about this choice, including consent withdrawal, please see our For information about this choice, including consent withdrawal, please see our Privacy Policy .

Reviewer #1: No

Reviewer #2: No

---

## [Author Response · Author response to Decision Letter 1]

4 Feb 2026

Thanks to The Editor and Reviewers,

Responses to Editor and Reviewers provided in word file name (Response to Reviewers)

---

## [Decision Letter · Decision Letter 1]

23 Feb 2026

Diet and density effects on growth, proximate composition and fatty-acid profile of Hediste diversicolor juveniles

PONE-D-25-64293R1

Dear Dr. Pajand,

We’re pleased to inform you that your manuscript has been judged scientifically suitable for publication and will be formally accepted for publication once it meets all outstanding technical requirements.

Kind regards,

Eman Zahran, Ph.D

Academic Editor

PLOS One

Additional Editor Comments (optional):

Dear Author,

We are pleased to inform you that your manuscript entitled “Diet and density effects on growth, proximate composition and fatty-acid profile of Hediste diversicolor juveniles” (Manuscript ID: PONE-D-25-64293R1) has been accepted for publication in PLOS One.

We appreciate your careful attention to the reviewers’ and editors’ comments. The revisions have satisfactorily addressed all concerns and have strengthened the manuscript’s quality and presentation.

Thank you for choosing PLOS One for the publication of your work.

Reviewers' comments:

Reviewer's Responses to Questions

**Comments to the Author**

Reviewer #1: All comments have been addressed

Reviewer #2: All comments have been addressed

2. Is the manuscript technically sound, and do the data support the conclusions?

Reviewer #1: Yes

Reviewer #2: Yes

3. Has the statistical analysis been performed appropriately and rigorously?

Reviewer #1: Yes

Reviewer #2: Yes

4. Have the authors made all data underlying the findings in their manuscript fully available?

Reviewer #1: Yes

Reviewer #2: Yes

5. Is the manuscript presented in an intelligible fashion and written in standard English?

Reviewer #1: Yes

Reviewer #2: Yes

Reviewer #1: The authors conducted each comments. Well done for corrections. I accepted with this version of R1 of the manuscript.

Reviewer #2: The authors responded appropriately to the questions raised and improved the manuscript in accordance with the suggestions made.

**Do you want your identity to be public for this peer review?** For information about this choice, including consent withdrawal, please see our For information about this choice, including consent withdrawal, please see our Privacy Policy .

Reviewer #1: No

Reviewer #2: No

<!--a=1<!--a=1

---

## [Editor Report · Acceptance letter]

PONE-D-25-64293R1

PLOS One

Dear Dr. Pajand,

I'm pleased to inform you that your manuscript has been deemed suitable for publication in PLOS One. Congratulations! Your manuscript is now being handed over to our production team.

Kind regards,

on behalf of

Professor Eman Zahran

Academic Editor

PLOS One